# Why Veterinarians (Do Not) Adhere to the Clinical Practice *Streptococcus suis* in Weaned Pigs Guideline: A Qualitative Study

**DOI:** 10.3390/antibiotics12020320

**Published:** 2023-02-03

**Authors:** Isaura Y. A. Wayop, Emely de Vet, Jaap A. Wagenaar, David C. Speksnijder

**Affiliations:** 1Department Biomolecular Health Sciences, Faculty of Veterinary Medicine, Utrecht University, Yalelaan 1, 3584 CL Utrecht, The Netherlands; 2Consumption and Healthy Lifestyles Group, Wageningen University and Research, 6700 EW Wageningen, The Netherlands; 3Wageningen Bioveterinary Research, Houtribweg 39, 8221 RA Lelystad, The Netherlands; 4University Farm Animal Clinic ULP, Reijerscopse Overgang 1, 3481 LZ Harmelen, The Netherlands

**Keywords:** veterinary guidelines, antimicrobial stewardship, qualitative research, *S. suis*, implementation strategy, Dutch pig industry, antimicrobial use, Dutch veterinarians, determinants, theoretical domains framework

## Abstract

The Netherlands has been very successful in the last decade in reducing antimicrobial use in animals. On about a quarter of farms, antimicrobial use in weaned pigs remains relatively high. As *Streptococcus suis* (*S. suis*) infections are responsible for a high consumption of antimicrobials, a specific veterinary guideline to control *S. suis* was developed, but seemed to be poorly adopted by veterinarians. Guided by the theoretical domains framework, the aim of this study was to identify determinants influencing veterinarians’ adherence to this guideline. We interviewed 13 pig veterinarians. Interviewees described multiple approaches to managing *S. suis* problems and adherence to the guideline. Mentioned determinants could be categorized into 12 theoretical domains. The following six domains were mentioned in all interviews: knowledge, skills, beliefs about capabilities, beliefs about consequences, social influences, and environmental context and resources. The insights derived from this study are relevant for understanding factors influencing veterinarians’ adoption of scientific evidence and guidelines and can be used to develop evidence-based implementation strategies for veterinary guidelines.

## 1. Introduction

Antimicrobial resistance (AMR) is considered a threat to human and animal health around the world. One of the objectives in the global action plan launched by the World Health Organization to combat the increasing threat of AMR is to optimize the use of antimicrobials in humans and animals [1]. To meet this objective, it is necessary to search for evidence-based and practical approaches to optimize antimicrobial use (AMU) in various animal production systems.

In the Netherlands, the use of antimicrobials in animals decreased impressively be-tween 2009 and 2021 by 70.8% after the introduction of various regulations and measures [2]. A wide variation still exists, however, among farmers in their level of AMU and among veterinarians in their antimicrobial prescription patterns [2,3]. Targeting higher users/prescribers offers a possibility for further AMU reduction.

One measure taken by the Royal Dutch Veterinary Association was the development of veterinary clinical practice guidelines. These guidelines contain evidence-based recommendations for the clinical approach to (e.g., diagnostics, treatment options) and the prevention of specific animal diseases associated with a substantial AMU. The veterinary guidelines can support veterinarians in their clinical decision-making, including antimicrobial prescribing practices [4]. These guidelines do not have a legal status, but they are part of a voluntary veterinary quality system. The guidelines aim to represent the applicable professional standard at the time of preparation within the context of laws and regulations. Veterinarians’ deviations from recommendations in the veterinary guidelines should be substantiated with good reasoning, because the guidelines are considered to describe the professional evidence-based state-of-the-art approach to common clinical conditions. The veterinary guidelines were published online, and veterinarians were made aware of them through newsletters and other communication channels. However, there was no comprehensive approach to support veterinarians’ adoption of these guide-lines.

The *Streptococcus suis (S. suis)* in weaned pigs guideline was one of the first of these veterinary guidelines developed and published in the Netherlands [5]. The reason for publishing the *S. suis* guideline was that *S. suis* infections are a major problem, there is no effective commercial *S. suis* vaccine available, and, as a result, antimicrobials are used extensively to control infections, reduce economic losses, and prevent mortality in weaned pigs [6,7]. The *S. suis* guideline includes well-considered recommendations about prevention measures, the use of auto-vaccines and treatment of clinical disease, the reporting and the evaluation of the treatment, and a flowchart with multiple decision points to support veterinarians to prescribe antimicrobials in a careful, selective, and responsible way. A survey in 2016, as part of a student thesis, among veterinary practitioners concerning the implementation of the *S. suis* guideline showed that a large majority of respondents used this guideline only partly (66%) or not at all (15%) in practice for various reasons [8]. The results of this survey were not translated into a revision.

Despite the positive impacts of clinical guidelines on supporting prudent antimicrobial prescribing in human medicine, studies have shown that recommendations in guide-lines are often not fully implemented in human practice. Various determinants can influence clinicians’ clinical decisions and have been studied extensively in different clinical settings [9,10]. Various reasons at the level of the guideline itself (e.g., complexity, procedural clarity), the strategy chosen to disseminate the guideline, varying attitudes of pro-posed users toward the guideline, and the medical professional’s working environment have an impact on how and to what extent guidelines are adopted (decision to work ac-cording to the guidelines) [11,12,13].

Evidence-based implementation strategies (comprehensive guideline introduction strategies based on scientific evidence) have been developed in human medicine after the careful assessment of obstacles to adoption and evidence-based theory. These implementation strategies, when introduced, have been shown to be effective in promoting the adoption of clinical guidelines in human medicine [9,14,15]. Evidence-based implementation strategies for veterinary guidelines are still lacking, however. This might explain Dutch pig veterinarians’ reported poor uptake of the *S. suis* guideline [5].

To develop an effective evidence-based implementation strategy for veterinary guidelines, insight is needed into the determinants that affect veterinary clinical decision-making. Theories, such as the theoretical domains framework (TDF), help to identify behavioral determinants for understanding complex behaviors [13]. The identified determinants can, in turn, inform the design of theory-based interventions to support the uptake of veterinary guidelines. A limited number of studies have explored determinants of clinical decision-making and antimicrobial prescribing practices by veterinarians in general [16,17,18,19]. However, no scientific study has yet investigated the specific determinants of veterinary practitioners’ adoption and implementation of veterinary guidelines. The aim of this study was to identify determinants influencing pig veterinarians’ adherence to the *S. suis* guideline.

## 2. Materials and Methods

### 2.1. Study Design

We used a qualitative research methodology with face-to-face interviews to explore Dutch pig veterinarians’ perceptions, adoption, and implementation of the *S. suis* guideline.

The TDF outlines theoretical determinants of behavior and was considered to provide an excellent framework to explore the determinants that influence veterinarians’ decision-making process regarding adherence to, and implementation of, the *S. suis* guideline [13]. The TDF was specifically developed and validated to advance the field of implementation research and particularly facilitates cross-disciplinary implementation research. The TDF is a synthesis of multiple existing psychological theories about the drivers of behavior and proposes determinants of implementation in overarching domains [13,14,20]. It has been successfully used in various clinical settings to develop theory-informed interventions to improve the implementation of guidelines in human medicine [20,21,22]

### 2.2. Instrument Development and Data Collection

The semi-structured interview guide that we developed to interview veterinarians was based on the outcomes of a survey conducted in 2016 among pig veterinarians regarding their perspectives on the *S. suis* guideline, on our previous experiences with qualitative interviews among veterinarians, and on interviews with authors of the *S. suis* guideline and with a European veterinary specialist in porcine health management [8,17]. A pig veterinarian was consulted to pilot the draft interview guide before we formulated the final version. The pig veterinarians involved in the interview development did not participate further in the study.

The interviews all started with an open question relating to participants’ experiences with *S. suis* and ended once all the topics from the interview guide, in different orders, were discussed. During the interview process, we maintained the balance between a comfortable respondent and receiving the information deemed necessary for a thorough, in-depth understanding of respondents in qualitative research [23].

The most important topics addressed were: *S. suis* infections (clinical signs, laboratory investigations conducted, antimicrobial use, the use of *S. suis* auto-vaccines and other preventive measures), the farmers’ influence on veterinarians’ clinical advice, the reporting of farm visits, the veterinarians’ view on and usage of the *S. suis* guideline, and more generally on their perceptions of veterinary guidelines. The first interviews were used to adapt the focus of subsequent interviews with minor changes to the interview guide as part of an iterative approach [16]. The duration of the interviews ranged between approximately 40 and 120 min. On average, the interviews lasted 75 min. The interview guide for the veterinarians is attached in Appendix A.

### 2.3. Participants

The Netherlands has a relatively large pig industry with approximately 11.5 million pigs on approximately 3000 pig farms that receive veterinary care from a total of approximately 225 registered pig veterinarians, many of whom are clustered in several large specialized group practices [24,25]. Pig veterinarians in the Netherlands were approached, on an ongoing basis, by phone and/or email with the request to participate in this study until theoretical saturation was achieved. The inclusion criterion for pig veterinarians was that they provided veterinary care for farms with weaned pigs. The veterinarians were invited based on a purposive selection in different geographical locations in the Netherlands and years of clinical experience. Thirteen veterinarians, representing 10 veterinary practices, were willing to participate in the interviews. Two veterinarians approached were not willing to participate in the interviews for unknown reasons. We included the largest veterinary practices in the Netherlands to ensure that the veterinary practices responsible for a substantial part of the veterinary care of Dutch pig farms were represented in the study. The participants were interviewed between October 2018 and May 2019. Table 1 shows the participating veterinarians’ basic demographic information.

The interviews were all conducted in Dutch by the author (IW, researcher and veterinarian). The author was trained by experienced interviewers (EdV, DS). After the first contact by phone or email, all participants received information about the study and an informed consent form. Prior to each interview, the informed consent was signed by the interviewer and the participant to guarantee that all data gathered in the interview would be handled confidentially and to underline the voluntary nature of participation. All interviews took place at a convenient location for the participant (veterinary practice, restaurant) to secure a comfortable environment, were audio recorded, pseudo-anonymized using personal identifiers, transcribed verbatim, and saved in a secured data storage system at Utrecht University (YODA).

### 2.4. Data Analysis

The interviews were coded on the basis of the transcripts and analyzed using NVivo 12 (version 12 Pro, Windows). The researcher (IW) used an iterative approach on an ongoing basis when analyzing the transcripts and repeated the analysis for the first five transcripts based on her experience and discussion with the other authors [16]. After 10 interview analyses, no new information appeared and theoretical saturation was reached. Three additional interviews were conducted with veterinarians to ensure that no new information could be found about barriers and facilitators influencing veterinarians’ adherence to the guideline.

Open coding (breaking data into discrete parts to create codes with labels) was chosen as a first step to ensure that no preconceived notions and biases were involved [14]. Another researcher (DS) analyzed and coded two randomly selected interviews independently to discuss the reliability of the interviewer’s (IW) coding process and analyses. The authors had several meetings to make sure that the open coding process was undertaken correctly. The open coding process resulted in 356 codes of determinants that are believed to influence veterinarians’ decision-making process regarding adherence to the *S. suis* guideline.

The second step in the coding process was to make sense of and synthesize the 356 codes into a set of higher-order theoretical determinants of behavior and map the determinants in the original domains of the TDF [13].

### 2.5. Data Triangulation with Farmers

To cross-validate the results of the interviews with the veterinarians, we interviewed farmers to elicit their views on the matters brought forward by the veterinarians. Nine pig farmers were approached by their veterinarians with a request to participate in this study. The selection criterion for the pig farmers was meeting the definition of an *S. suis* problem farm as defined in the *S. suis* guideline (use of second-choice antimicrobials and/or an AMU above the nationally defined threshold value to treat/control *S. suis* infections). Five farmers approached were willing to participate in the interviews. Four farmers approached were not willing to participate in the interviews for unknown reasons. The interview guide for the farmers was developed based on the results of the interviews with the veterinarians. The interview guide for the farmers also contained questions about the relationship with, and expectations of, their veterinarian. The interview guide for the farmers is attached in Appendix A. Table 2 shows the participating farmers’ basic demographic information.

## 3. Results

The results gave an overview of determinants influencing veterinarians’ conscious and unconscious decisions about adhering to the *S. suis* guideline. The given answers in this study appeared to be open and unbiased. Determinants were divided into 12 domains: knowledge, skills, beliefs in capabilities, beliefs about consequences, motivation and goals, memory, attention and decision processes, nature of the behaviors, social/professional role and identity, emotion, social influences, and environmental context and resources of the TDF [13]. We did not map any codes in the domain behavioral regulation of the TDF [13]. An overview of the theoretical determinants and their constructs are presented in Table 3.

The results indicated that six domains were mentioned consistently in all interviews with veterinarians (knowledge, skills, beliefs about capabilities, beliefs about consequences, social influences, and environmental context and resources), whereas the other domains (motivation and goals, memory, attention and decision processes, nature of the behaviors, social/professional role and identity, and emotion) were covered more incidentally during the interviews. This may indicate that these six consistent domains are relatively more important for guideline adherence. In the following, the six consistently reported domains are presented. Examples of the other domains identified are provided in Appendix B.

### 3.1. Knowledge

Determinants of adherence to the *S. suis* guideline included knowledge of the recommendations/content of the *S. suis* guideline itself, theoretical veterinary knowledge (e.g., about *S. suis* auto-vaccines, effects of antimicrobials on the gut of weaned pigs), knowledge on regulations and laws related to AMU, and knowledge about advisory techniques.

In the Netherlands, veterinary antimicrobials are classified as first-, second-, and third-choice antimicrobials, where first-choice antimicrobials can be prescribed empirically and third-choice antimicrobials can be prescribed to individual animals only after susceptibility testing because of their importance for public health [26]. The *S. suis* guideline states that an *S. suis* problem farm is a farm where the use of antimicrobials for treating weaned pigs with clinical symptoms of *S. suis* results in a level of AMU above the threshold value (at the moment 20) and/or the use of second-choice antimicrobials (following the classification of veterinary AMU according to Dutch policy). Many veterinarians did not know exactly when a farm should be regarded as an *S. suis* problem farm, although a problem farm requires additional veterinary attention and diagnostic approaches according to the *S. suis* guideline. Examples of participants’ answers if asked *“what is your definition of a problem farm”* include: *“If I have to give medication structurally. And that are also farms with a high AMU, above the threshold of 20 DDDA [Defined Daily Dose Animal], and where also the mortality is above 2.5%”* (P3)*;*
*“I think that it is not a problem farm if the DDDA is under 20. And I think below 50 either, if I have to say. If it is above 50 then you have to ask yourself, am I doing it right?”* (P12).

A recommendation in the *S. suis* guideline is to start with individual antimicrobial treatment of weaned pigs with clinical symptoms caused by *S. suis* until there are 5% or 4% diseased weaned pigs within a group in 5 days or 24 h, respectively, before initiating metaphylactic group therapy. The majority of the respondents were not able to recall these recommended threshold values. Multiple respondents believed that the threshold value referred to the percentage of dead weaned pigs, whereas the threshold value in the guideline actually refers to sick weaned pigs. For example: *“the percentage mortality for application (of group treatments), to me they were quite... I do not know exactly what it was?”* (P4).

If participants were asked about the issues with which they (dis)agreed in the *S. suis* guideline recommendations, some respondents indicated that they had to re-read it because they could not remember the specific content: *“Then I have to read it again, I cannot remember the guideline clearly”* (P 13).

### 3.2. Skills

The respondents described a range of skills that, according to them, are required to be able to completely implement the *S. suis* guideline. The guideline states that the *S. suis* approach needs to be reported in farm visit reports and gives statements that need be followed up with the farmer. Reports can be necessary for a complete evaluation of the *S. suis* approach. The most important skills mentioned were: the ability to produce complete and concise farm visit reports in the limited time period that veterinarians experience in practice; proper communication skills; building trust in the relationship with farmers; advisory and teaching skills; and being able to make a professional decision under pressure of time, regulations, and/or clients’ conflicting opinions: *“yes, it is my job to educate them* [farmer and his employees] *well”* (P2); *“Often we have the right skills to solve the problem, but sometimes it takes time, but we have those skills to treat the animals but also* [educate] *the farmer”* (P10).

Making complete farm visit reports that meet the requirements as set in governmental regulations and private quality regulations in a limited amount of time was considered to be difficult for some participants: *“I try to do so as much as possible, but that will sometimes fail, yes. But that is also because…, I am probably not a paper person”* (P11). Other less-experienced participants mentioned that making complete farm visit reports was easy for them.

### 3.3. Beliefs about Consequences

The participants expressed diverging opinions about the effects of veterinary guidelines on clinical outcomes, antimicrobial use, veterinarians’ professional autonomy, and other parameters. The same holds true for the perceived or experienced effects of several parts of the *S. suis* guideline specifically.

The participants with less clinical experience indicated that parts of the *S. suis* guideline, or veterinary guidelines in general, were useful for them: *“I think guidelines are good for young veterinarians, the starters. And for myself, sometimes I want to standardize certain procedures in our veterinary practice”* (P3); *“I think, in itself, it is good that the guideline exists, so the veterinarians know what is expected of them, they can rely on them. Also, for the farmers, it makes it easier”* (P7). Other veterinarians did not consider the *S. suis* guideline or other veterinary guidelines useful for them and were not convinced that adherence to the veterinary guidelines would lead to positive results at the farm or a decrease in AMU: *“The guidelines contain a lot of words, but for a practitioner, for problem farms, the S. suis guideline does not bring me further”* (P8); *“My first thought was that making the guideline was a waste of time, paperwork without value. The subject was the decline of antimicrobial use. And that happened before this kind of veterinary guidelines existed. The guideline provides frustration for veterinarians”* (P1).

Some participants believed that the *S. suis* guideline did not result in better health and welfare for the weaned pigs or in less AMU. The majority of the participants indicated that the thresholds to shift from individual to group treatments were too high for some farms. A negative effect on the weaned pigs’ health and welfare was expected if this recommendation of the *S. suis* guideline was strictly followed. The veterinarians differed clearly on which antimicrobial class (first- or second-choice) they preferred to prescribe in the event of an *S. suis* infection and whether they recommended that farmers administer the antimicrobials to individual weaned pigs (parenterally) or as a group treatment (orally via feed or water). This difference was also observed among colleagues who worked in the same veterinary practice.

*“I think 90% of my farmers use Procaïne-benzylpenicilline* [a first-choice antimicrobial applied parenterally]” (P4); *“My first advice is amoxicillin* [second-choice antimicrobial applied orally as group treatment]” (P1); *“Procaïne-benzylpenicilline* [first-choice] *works well for* [weaned pigs] *with meningitis, if the* [weaned pigs] *are not too far gone”* (P6). The two most frequently mentioned reasons for deviating from the threshold to initiate group treatments in the *S. suis* guideline were (i) the dubiety of the effectiveness of first-choice parenterally applied antimicrobials (e.g., presence of antimicrobial-resistant *S. suis* strains, the effectiveness of the dosage when applied as stated in the summary of product characteristics, and the perceived small time window between first symptoms and the curative treatment in order to be effective) and (ii) the belief that there are no negative consequences of treating groups instead of treating individuals with antimicrobials. The most frequently mentioned reasons for not initiating group treatment with amoxicillin and, thus, following the AMU recommendations of the *S. suis* guideline were the believed long-term negative side effects in the treated pigs (e.g., reduced resilience of piglets to infections) of oral amoxicillin application and the estimation that a farmer had sufficient management skills that reduce the necessity of antimicrobial treatments overall: *“with amoxicillin* [second-choice], *I think you jump from the frying pan into the fire”* (P4); *amoxicillin is effective, only there are side effects, in their intestines, especially for young animals* (P12).

### 3.4. Beliefs about Capabilities

The participants talked differently about their self-esteem and optimism/pessimism about resolving problems caused by *S. suis* on pig farms with or without the use of the *S. suis* guideline.

Respondents reported that they did not believe that it was always possible to alleviate *S*. *suis* problems given the current pig husbandry system (see environmental context and resources). An important reason for a lack of confidence was having negative experiences with attempts to resolve *S. suis* problems where they, despite a lot of effort, did not succeed: *“despite everything we have done, we still have an antimicrobial use above the threshold, and he* [farmer] *still has outbreaks. You are checking the* [weaned pigs] *and nothing is wrong, 30 min later, 15* [weaned pigs] *recumbent, very frustrating”* (P4); *“I am not able to solve S. suis problems structurally on the farms”* (P3). Other respondents answered that they had a lot of confidence in their problem-solving capabilities and their ability to find the bottlenecks to control *S. suis* problems: *“Yes, I’m that arrogant, I don’t need a guideline, I solved it without it before”* (P2). The respondents did not mention that they felt incapable of following the recommendations of the *S. suis* guideline.

### 3.5. Social Influences (Norms)

The veterinarians stated that the decision (not) to follow the *S. suis* guideline was, to some extent, influenced by the farmers, their colleagues, and other farm advisors such as nutritionists.

When respondents were asked how they approached an *S. suis* problem, the majority recalled seriously taking into account their pig farmer’s experience and opinion in their advice: *“In fact, we can diagnose the disease quite fast and we also take the experience of the farmer into account”* (P9). Veterinary participants all mentioned that their role was to advise the farmers but ultimately the farmers made the decision.

Veterinarians mentioned that colleagues’ opinions had a huge influence on their decisions about how they handled an *S. suis* problem on a farm and the positive effect of having experienced and knowledgeable colleagues and structural education: *“I think colleagues, a big team and peer-consultations, they are more valuable than any guideline”* (P10).

### 3.6. Environmental Context and Resources

All veterinary participants described multiple important external factors that influenced adherence to the *S. suis* guideline but which they could not always influence. A few frequently mentioned examples included the multifactorial nature of *S. suis* outbreaks in pigs, the composition and quality of the feed for the pigs, existing laws and regulations of the government and private quality systems, veterinary practice policy/protocols, and the pigs’ housing conditions.

Nearly all veterinary respondents described the huge influence of a ration of suboptimal feed quality or composition on the burden of *S. suis* problems on farms. Veterinarians mentioned that they often felt that they were not in a position to influence the composition and quality of the feed; they felt dependent on sometimes conflicting advice from feed advisors and were not always able to find information on the specific composition of the feed to judge its quality: *“Feed advisors can give any advice without having any responsibility for the consequences”* (P2); *“I want more information from the feed advisors, but they don’t always tell me everything”* (P5). Maintaining a good relationship with feed advisors was mentioned as beneficial for resolving *S. suis* infections as it provided better insights into the weaned pigs’ nutrition and the ability to change the ration.

Some participants did not support the existing laws and regulations of the government and private quality systems, including veterinary guidelines. They regarded them as unclear, not practical, and sometimes conflicting: *“the rules are unclear, there are different laws that are not all consistent, different guidelines in countries, there is a lot, but they all say something different”* (P13). Some participants felt that their professional autonomy was limited under the influence of laws, regulations, and guidelines that they did not support.

In some veterinary practices, the existing practice policy had an influence on adherence to the *S. suis* guideline. One recommendation in the *S. suis* guideline was to conduct a post-mortem examination of weaned pigs at least twice or four times per year. Veterinarians from veterinary practices that followed this advice mentioned that giving reminders to the farmers was part of their normal routine and this was implemented by a majority of the farms: *“we have a practice policy for problem farms, 4–5 times per year they get a reminder to send in* [weaned pigs] *for post-mortem examination, this is additional to the farm visits”* (P5).

### 3.7. Data Triangulation with Farmers

During the interviews with farmers of *S. suis* problem farms, it appeared that they were all satisfied with their veterinarian’s approach to controlling *S. suis* infections. They expressed their confidence and trust in their veterinarian and blamed the complex, multifactorial nature of *S. suis* infections for the lack of results: *“the result is not as it should be, but I don’t think that I can blame my veterinarian”* (P17).

Consistent with the veterinarians, the farmers were of the opinion that they make the ultimate management decisions and the veterinarian is their advisor when it comes to animal health: *“He* [his veterinarian] *advises, he says watch that or that, yes, and what we do with the advice is ours”* (P17).

We also found that some views of the veterinarians contradicted the specifically expressed opinions of the farmers, which indicates a misperception on the veterinarian’s part. As an example, one veterinarian held the perception that all farmers have a preference for *amoxicillin* to treat their pigs. However, there appeared to be different views on the pros and cons of using *amoxicillin:*
*“Amoxicillin is such trash, yes. You devastate the inside of a pig”* (P14); *“that TMPS* [first-choice antimicrobial for group treatment] *decreases the pressure but is not as effective as amoxicillin, that* [amoxicillin] *works well”* (P18). Another example pertains to the perceived threshold at which a farmer will contact a veterinarian and shift from individual to group treatment. Compared to their veterinarians, farmers appeared to accept a higher threshold before shifting to group treatment: *“I follow that threshold* [4% in 24 h], *but 5% in 5 days, I think that is way too long, S. suis is persistent and sometimes fast, you cannot control the mortality anymore. With S. suis, in my opinion, you have to react fast”* (P15); *“if the mortality is 6%, then I need amoxicillin”* (P17). Furthermore, there were misperceptions on the value of the farm visit reports. While some veterinarians in our study believed the farmers did not value or even read the reports and regarded it as an unnecessary formal exercise, all farmers expressed that farm visit reports had a value for their farm: *“In itself, they* [the reports] *are good, because from time to time I look back on them”* (P16); *“Yes, that* [report] *is valuable information, you can always look back”* (P15).

## 4. Discussion

Guideline development, implementation, and uptake.

Clinical practice guidelines describe the professional evidence-based state-of-the-art approach and are developed to support the clinician. Clinical practice guidelines are not a mandatory protocol that must be followed blindly by a clinician. However, some participants expressed negative perceptions regarding the veterinary guidelines in general. They feared losing their autonomy to act as veterinary professionals and were concerned that strict adherence to the guidelines could result in negative outcomes for patients.

The introduction of clinical guidelines in veterinary medicine in the Netherlands is a recent development, in contrast to the clinical guideline development in human medicine, which started in the 1970s. When clinical guidelines were introduced in human medicine, primary care physicians had comparable negative perceptions [27]. When general practitioners in the Netherlands started to work with clinical practice guidelines, adherence was low, whereas for the newer generations of general practitioners, 100 new guidelines and 23 years later, the clinical practice guidelines are part of their (continuing) education and assessments and currently seen as the professional clinical standard [28,29,30]. These guidelines are structurally updated and continuously evaluated through feedback given by healthcare professionals [30,31]. These guideline developers still face challenges (e.g., mechanisms to update guidelines, collaborative initiatives to avoid contradictory recommendations), but research shows that primary care physicians, in general, value being supported in their clinical work by guidelines that have been shown to improve the quality of care for patients [32,33,34,35].

In veterinary medicine, clinical practice guidelines can also play an effective role in making decisions easily and preventing mistakes, as has been described for small animal clinics [33]. Veterinarians often welcome such clinical practice guidelines and they use them to improve animal care and justify their veterinary decisions to clients, although these guidelines are not specifically about AMU and we do not have evidence for the effect of the veterinary guidelines in the Netherlands, therefore making it not fully comparable.

Clinical decision-making, like the decision (not) to prescribe antimicrobials, is complex, as is recognized in both human and veterinary medicine [9,10,16,17]. The development of veterinary guidelines in the Netherlands was part of a broader strategy to reduce AMU in animals. When the veterinary guidelines started to be published, a substantial reduction in AMU, approximately 60% since 2009, had already taken place thanks to harvesting the low hanging fruit, such as abandoning routine prophylactic AMU practices and shortening therapy durations [2,4]. Respondents to our study mentioned that the veterinary guidelines did not have an effect on the AMU. Although this can be argued, it is very possible that many of the recommendations in the guidelines were already in practice and implemented by veterinarians and farmers. In contrast to the human clinical guidelines, until now, the veterinary guidelines have been published without complete implementation strategies and a planned cycle of continuous improvement including evaluation and feedback. No financial structure was foreseen for revision and continuous updating.

### 4.1. The S. suis Guideline

#### 4.1.1. Knowledge

All participants were aware of the existence of the *S. suis* guideline, but the level of knowledge of its content varied markedly. Some participants did not know what the most important definitions in the guideline were, making it impossible to use the guideline correctly. It should, however, be noted that this guideline is extensive and contains many recommendations on a complicated clinical problem, making it difficult to remember all the details. A solution could be to continuously educate veterinarians on the *S. suis* guideline, for example via physical or online learning modules as part of post-graduate education, using an app with a summary of the *S. suis* guideline, or issuing reminders that become part of daily practice. Education is often part of interventions for the implementation of clinical practice guidelines in human medicine and, thus, could be considered in veterinary medicine as well [36,37].

#### 4.1.2. Skills

All the participants were convinced of their ability to diagnose *S. suis* correctly, but they described huge differences in administrative and communication skills that influence adherence to the *S. suis* guideline. The importance of proper communication skills and building a relationship with the farmer and other stakeholders and their effect on the prescription behavior of antimicrobials of veterinarians have also been observed in other studies [3]. Although the veterinarians in our study mentioned that they had sufficient advisory skills, studies tell us that veterinarians can improve their advisory skills despite their own judgement [38,39]. To improve veterinarians’ advisory skills, training can be advised.

#### 4.1.3. Beliefs about Consequences

The main objective of the *S. suis* guideline is careful, selective, and responsible use of antimicrobials for the clinical problems resulting from *S. suis* infections in weaned pigs. Some participants believed that implementing the *S. suis* guideline did not have positive consequences (better health, improved animal welfare, or less AMU) and mentioned specific recommendations in the guideline whose positive outcomes they doubted. Overall, there is evidence that, in general, the antimicrobial reduction policy in the Netherlands did not negatively affect the economic and technical performances of pig farms, but that does not rule out potential negative outcomes of specific measures [40].

Two frequently mentioned examples in our results, and in the results of the 2016 survey, were: (i) the threshold to shift from individual to group treatments was perceived as too high, resulting in higher morbidity and mortality rates, and (ii) the recommendation to use the first-choice antimicrobial procaine-benzylpenicillin was disputed because respondents experienced bad therapy outcomes despite susceptibility shown by antibiograms. These two statements can be seen as a huge impediment to the implementation of the *S. suis* guideline. This makes it a challenge for the developers of the veterinary guidelines to reach consensus about the guideline development process and the correctness of the recommendations in the guidelines in order to achieve a sufficient support base.

Our results indicate that veterinarians who had negative experiences with inspection authorities (e.g., prosecution by the Food Safety Authority) for adherence to AMU policies tended to be negative towards the veterinary guidelines in general. These veterinarians felt that veterinary guidelines could be used as a sanctioning framework by inspection authorities that would have a negative influence on their autonomy as a veterinary professional. On the other hand, our results indicated that veterinarians with positive experiences of using the guideline (e.g., use of checklist management measures in the guideline) tended to be more positive towards veterinary guidelines in general. This is also shown for attitudes toward AMU, as veterinarians who prescribe less antimicrobials tend to have more optimistic expectations of antimicrobial reduction for animal health, animal welfare, and public health compared to veterinarians who prescribe more antimicrobials [3]. This shows how both positive and negative experiences can influence veterinarians’ attitudes towards clinical guidelines in opposing directions. A theoretically appropriate intervention to support guideline uptake could include sharing case studies from colleagues who have had positive experiences after following the *S. suis* guideline in order to address and challenge the beliefs of veterinarians who anticipate negative outcomes.

#### 4.1.4. Beliefs about Capabilities

Our observations suggest that veterinarians with longer clinical experience and/or more guidance from colleagues (e.g., peer-learning) or who had received more education (e.g., coaching program) perceived more confidence for resolving clinical problems; this has also been seen in other studies [17]. These participants were more convinced that they could strongly direct the management decisions being made by farmers and felt strongly that they were in control of their work. This confidence could potentially positively influence the veterinarians’ enthusiasm for trying to convince or educate farmers on prevention regarding *S. suis* problems and adherence to the *S. suis* guideline when they are motivated to do so. We also observed how confidence in their own abilities to control *S. suis* infections was greatly dependent on their personal experiences with controlling *S. suis* infections.

The results of our qualitative study seem to indicate that more recently qualified veterinarians experience fewer problems with administrative tasks compared to more experienced veterinarians. Other studies agree that more recently qualified veterinarians who feel less confident to act as independent professionals might benefit from the support of guidelines and are more familiar with working with clinical guidelines than more experienced veterinarians [16,41]. A possible explanation could be that the more recently qualified veterinarians see the use of policies and guidelines on AMU as a quality standard for the veterinary profession and feel more confident with the administrative tasks because policies and guidelines are currently part of their education.

#### 4.1.5. Social Influences

The participants all agreed that social determinants could influence their adherence to the *S. suis* guideline. During the interviews, participants stated that these determinants were sometimes the reason why recommendations in the *S. suis* guideline could not be followed by the farmer despite veterinary advice. Some felt that certain social determinants for extensive antimicrobial use could be beyond their control, an observation that has also been described elsewhere [3]. Examples include the character of the farmer and influences of employees or family members. However, it is known that farmers identify their veterinarians as the most used, trusted, and independent source of information [42,43]. Therefore, proactively advising the farmer may (indirectly) contribute to better conditions to prevent *S. suis,* even when farmers do not request such advice.

Another veterinary guideline in the Netherlands, published around the same time, on the selective use of antimicrobials in dairy cows at drying off (instead of the blanket dry cow treatment that had been a routine for decades) was received and implemented well by veterinarians and farmers [44,45]. The selective use of antimicrobials in dairy cows at drying off guideline was spread not only to veterinarians but also to other stakeholders (farmers, feed advisors, dairy processors). The criteria for selective dry cow therapy were actively communicated by veterinarians and other stakeholders (e.g., via professional journals) as the new standard for dry cow therapy that met the newly introduced legal requirements that prohibited the preventive use of antimicrobials. This implies that, for better implementation, veterinary guidelines should be developed, disseminated, and, where necessary, enforced by all the relevant stakeholders in the sector. For the *S. suis* guideline, where the farmer’s role is very important in its implementation, it could be beneficial if other stakeholders in the pig industry (e.g., feed companies, piglet buyers) also actively supported and disseminated this guideline.

#### 4.1.6. Environmental Context and Resources

Our results show that environmental determinants play a big role in veterinarians’ adherence to the *S. suis* guideline. Examples include lack of financial funds, pinching laws and municipal regulations or licenses, characteristics of the existing housing that are suboptimal but cannot be changed overnight, or the piglets’ ‘quality’ (immune status). Although these structural factors cannot be influenced directly, it might be feasible to influence them indirectly.

Our results suggest that embedding guidelines in the veterinary practice’s policy influences veterinarians’ adherence to the *S. suis* guideline. On the other hand, veterinarians working in the same practice sometimes had different attitudes about adherence to the *S. suis* guideline. Therefore, we cannot conclude that all veterinarians will automatically follow the veterinary practice’s policy. Peer consultation groups could be beneficial to align the approach to *S. suis* problems applied by veterinarians in the same practice. All our study participants were positive about sharing knowledge with colleagues, but not all of them felt that they had the opportunity to do this (regularly).

In human medicine, regular peer consultation groups have been shown to be beneficial for reducing prescribing rates while maintaining satisfaction among patients. In veterinary medicine, veterinarians are described as experiencing multiple benefits of continuous education in veterinary peer study groups [46]. The results of our study and the experience from human medicine suggest that regular peer consultation meetings with colleagues in veterinary medicine could also help to improve the approach to *S. suis* problems and adherence to the *S. suis* guideline.

### 4.2. Implementation Strategy

One option to improve veterinarians’ adherence to the *S. suis* guideline is to develop a comprehensive implementation strategy built on theory and evidence, as described for clinical practice guidelines in human medicine [47,48,49]. To develop a sound implementation strategy, it is important to use a theoretical framework to choose the right behavior change techniques for these determinants in a practical way (real-world settings). Examples of behavior change techniques for such an implementation strategy can provide information, self-monitoring, or social support. Additional research is required to design and evaluate the effects of such a theory-based implementation strategy on the implementation of veterinary guidelines such as the *S. suis* guideline.

### 4.3. Strengths and Limitations of The Study

We performed 13 interviews and achieved data saturation; this is comparable to previous studies and reasonable given the overall population size (225 registered pig veterinarians) [50]. Because veterinary care and the laws and regulations can be different between countries, our results can probably not simply be extrapolated to other countries without adaptation to local differences.

The results obtained in our study provide an overview of determinants of why veterinarians (do not) adhere to the *S. suis* guideline. However, this study does not provide information about the importance of each of the determinants. We made a selection of six theoretical determinants that came up in all interviews. This does not mean that the determinants of other theoretical determinants are less important. For example, the veterinarians who are motivated to make good reports for the farmers, instead of complying with rules and laws, have a higher adherence to the *S. suis* guideline. We cannot tell how important this determinant is. To decide the relative importance of the different determinants, quantitative research is needed.

To structure our results, we used the TDF, the application of which is new in veterinary medicine and which was developed to identify determinants of behavior [14]. The TDF appeared to be a useful tool to structure the results of the interviews. Some expressions in the interviews related to more than one domain, in which case we chose to map that expression in the domain which fitted best, given the context in which the expression was made. For example, we mapped determinants involving the subject ‘peer-learning’ to the domain of environmental context and resources, as we see peer-learning in our study as education, which is a resource. You could also map determinants involving the subject ‘peer-learning’ to the domain of social influences, as colleagues may influence each other during peer-learning.

Our study also explored the usefulness of the TDF for enhancing the understanding of the initial implementation of veterinary guidelines. This step made it easier to differentiate the determinants, thereby making it a good start for the development of an implementation strategy for the *S. suis* guideline.

## 5. Conclusions

In this study, we identified determinants influencing veterinarians’ adherence to the *S. suis* guideline using a theory-based framework to understand psychological, social, and environmental determinants of behavior, i.e., the TDF. All the respondents mentioned factors related to the domains of knowledge, skills, beliefs about capabilities, beliefs about consequences, social influences, and environmental context and resources that influence their level of adherence to the *S. suis* guideline. Some of the respondents mentioned factors related to the domains of motivation and goals, memory, attention and decision processes, nature of the behaviors, social/professional role, and identity and emotion that influence their level of adherence to the *S. suis* guideline. Our results provide the basis for identifying behavior change techniques to enable a better uptake of the *S. suis* guideline.

## Figures and Tables

**Table 1 antibiotics-12-00320-t001:** Details of participants (veterinarians) interviewed in the study: “Why Veterinarians (Do Not) Adhere to the Clinical Practice Streptococcus suis in Weaned Pigs Guideline: A Qualitative Study”.

Interviewee	Current Role	Graduation Year Veterinarian	Years of Experience As PV	Number of Pig Veterinarians in VP or Partnership
1	Full-time PV	1990–1995	>15	>10
2	Full-time PV	1990–1995	>15	>10
3	Full-time PV	2010–2015	1–5	5–10
4	Part-time PV	2005–2010	1–5	>10
5	Full-time PV	1995–2000	>15	>10
6	Part-time PV	2005–2010	5–15	5–10
7	Part-time PV	1995–2000	>15	1–5
8	Part-time PV	1995–2000	>15	>10
9	Full-time PV	2000–2005	1–5	>10
10	Full-time PV	2010–2015	1–5	>10
11	Part-time PV	1995–2000	>15	1–5
12	Full-time PV	1990–1995	>15	1–5
13	Full-time PV	2000–2005	>15	5–10

PV = pig veterinarian, full-time ≥ 36 h weekly only pigs, part-time ≤ 36 h weekly only pigs, VP = veterinary practice.

**Table 2 antibiotics-12-00320-t002:** Details of participants (farmers) interviewed in the study: “Why Veterinarians (Do Not) Adhere to the Clinical Practice Streptococcus suis in Weaned Pigs Guideline: A Qualitative Study”.

Interviewee	Current Role	Number of Sows on The Farm	Years of Experience As PF	Number of Pig Veterinarians in VP or Partnership
14	Full-time PF	200–300	>15	>10
15	Full-time PF	500–600	>15	>10
16	Full-time PF	200–300	>15	5–10
17	Full-time PF	400–500	>15	>10
18	Part-time PF	600–700	5–10	5–10

PF = pig farmer, full-time = PF 100%, part-time = other job besides PF, VP = veterinary practice.

**Table 3 antibiotics-12-00320-t003:** Domains of the theoretical domains framework and what they construct in the study: “Why Veterinarians (Do Not) Adhere to the Clinical Practice Streptococcus suis in Weaned Pigs Guideline: A Qualitative Study”.

Domain	Constructs
Knowledge	The veterinarian’s knowledge regarding the *S. suis* guideline and handling *S.suis* problems.
Skills	The veterinarian’s skills/competence/ability regarding implementing the *S. suis* guideline.
Beliefs about capabilities	The veterinarian’s self-efficacy regarding implementing the *S. suis* guideline, including self-confidence/professional confidence, self-esteem, and optimism/pessimism.
Beliefs about consequences	The veterinarian’s anticipated outcomes, consequences, attitudes, and rewards regarding the *S. suis* guideline.
Motivation and goals	The veterinarian’s intention, intrinsic motivations, and goals regarding implementing the *S. suis* guideline.
Memory, attention, and decision processes	How easily the veterinarian remembers the (content of the) *S. suis* guideline and whether the veterinarian receives reminders about the guideline during the decision processes regarding handling *S. suis* problems at farms.
Nature of the behaviors	The veterinarian’s routine, automatic behavior, and habits regarding actions involved with the *S. suis* guideline.
Social/professional role and identity	The veterinarian’s professional identity and social group norms regarding the *S. suis* guideline.
Emotion	The veterinarian’s stress or frustration regarding the *S. suis* guideline.
Social influences	The veterinarian’s social support and group norms regarding the *S. suis* guideline. These include the opinions and behaviors of colleagues and the farmer.
Environmental context and resources	The veterinarian’s environmental constraints and resources/material resources (availability and management) regarding the *S. suis* guideline. These include the veterinary practice’s policy, the farm layout, laws and regulations, other stakeholders and advisors, etc.

Six domains were discussed in all 13 interviews with veterinarians: knowledge, skills, beliefs about capabilities, beliefs about consequences, social influences, and environmental context and resources.

## Data Availability

Not applicable.

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
