# Peer review of "Why Veterinarians (Do Not) Adhere to the Clinical Practice Streptococcus suis in Weaned Pigs Guideline: A Qualitative Study"

_antibiotics, 2023, doi:10.3390/antibiotics12020320_

Round 1
Reviewer 1 Report
The main question addressed by the research is the willingness to accept the guidelines of the Clinical practice of S. suis in weaned pigs by veterinarians and the observance of guidelines for veterinarians in practice. The research was carried out by extensive and detailed questioning of veterinarians and farmers in the form of interviews with predetermined questions, the same for all respondents, in several main areas of discussion (knowledge, skills, beliefs in capabilities, beliefs about consequences, motivation and goals, memory, attention and decision processes, nature of behavior, social/professional role and identity, emotion regulation, and misperception). The questions of the survey is very topical, especially from the point of view of the reason for proposing other treatment guidelines in general. Not only because of the incorporation of the latest scientific knowledge into these recommendations, but in this particular case mainly because of the expected of consequence of following this guideline - reducing the use of antibiotics and limiting the development and spread of resistant bacterial populations to antibiotics. The survey conducted provides guidelines makers with excellent feedback and opinions from practice, how easy or difficult it is to follow or trust various recommendations, and what is the willingness of veterinary practitioners in the field to change or modify their practices. Feedback on the implementation of the "theoretical" recommendations in practice is invaluable, and therefore I consider the publication of the submitted survey results very important, beneficial and informative. Methodologically, the survey is well developed, the questions are appropriately chosen and the answers have been analyzed in a sophisticated way using the program NVivo 12 (version 12 Pro, Windows). The conclusions are drawn realistically, showing that established practices that have been used for many years are still preferred among some practicing veterinarians and only partially is followed as recommended in the guidelines. This points to the fact how important it is to motivate veterinarians to comply with the issued guidelines, for example in the form of a presentation by the authors to defend the issued recommendations to the professional public at various educational and informational events. References are appropriately chosen and correspond to the text and content of the manuscript. The manuscript contains three Tables and two Appendices with another three tables. The tables are clear and well-designed. I believe that the topic of the survey is very current and the results interesting enough that I recommend the manuscript for publication in the journal Antibiotics, in its original form and without modifications.Author Response
We thank Reviewer 1 for his/her efforts to review our manuscript and for the very positive recommendation.
Reviewer 2 Report
Review of “Why veterinarians (do not) adhere to the clinical practice Streptococcus suis in weaned pigs guideline: a qualitative study”
Overall Comments
This manuscript reports an interesting study exploring different factors (determinants) that influence a specific behaviour by pig veterinarians – whether or not they decide to adhere to clinical practice guidelines for treating S. suis in pigs. One potential strength of this study is the use of a psychological framework – the Theoretical Domains Framework – to explore the determinants of this behaviour. However, in its current draft I have questions relating to the methods and results, which require further information (and potentially further analysis) to be presented.
My recommendation is major revisions. The essence of the study is interesting and could potentially make an important contribution (by highlighting the importance of using psychological/behavioural science frameworks to gain deeper insights into the wide range of factors that can influence veterinarians’ decisions to follow, or not follow, clinical guidelines). To strengthen the manuscript for publication standard, I would like to see more information presented by the authors about the specific process of coding and analysis, and I would also like to see more detail presented within the results. I also have questions about the appropriateness of including the farmer interviews in the dataset for this study. I have set out specific comments below.
Abstract
Well-written and covers the key points from the study.
Introduction
The introduction is well-structured and well-written, providing a concise overview of the problem and justification for the study. I have one suggestion for strengthening the introduction, which is to include a brief mention of the TDF and a high-level justification for using it, e.g., that prescribing and treatment behaviours by vets are complex, and we need to use theory-based psychological/behavioural models to develop a greater theoretical understanding of the range of factors (determinants) that can influence prescribing/treatment behaviours. Once we understand which factors are important in driving a specific behaviour (i.e. adhering to guidelines) then we can develop theory-based behaviour change interventions to target these factors.
Methods
Although there is precedence in the AMR literature for combining vet and farmer datasets in qualitative studies for analysis, it is not clear to me why the 5 farmer interviews have been included in with the vet interviews for this particular study. Based on the current draft of the manuscript, there has not been a sufficient rationale presented to justify the merging of these two datasets, and my current recommendation is that the farmer interviews should be excluded from the dataset (unless the authors can present a robust justification for inclusion). The research question is framed around what influences vets to (not) follow the guidelines – and based on the interview schedules, it does not appear that farmers were really being asked about why they thought vets did/did not follow the guidelines, but more about their own practices and decisions as farmers. I would request the authors to either:
· Provide more detail to justify the inclusion of the farmer interviews with the vet dataset – and if it is only to support/challenge what the vets said and the vets’ perceptions (as per lines 177-178), then this needs to be made much clearer in the methods and results (for example, in the results, any findings made from the farmer dataset need to be very clearly stated, including which TDF domains were coded for in the farmer interviews, which were coded for in the vet interviews, and whether quotations come from vets or farmers). If they choose to retain the farmer interviews, I would also ask the authors to consider separating out the farmer analysis, perhaps presenting it briefly at the end of the results, highlighting where the farmer comments supported or challenged vets’ perceptions of the farmers. However, with only 5 farmers, I would suggest that this might be too small a sample to draw conclusions from – and exploring whether there are misperceptions between vets and farmers is really a separate research question and not the main focus of this paper
· Alternatively, the authors could remove the farmer interviews from this analysis/manuscript – I am not clear what the farmer analysis ‘adds’ to the analysis and interpretation of the vet data. Qualitative methods allow for much smaller samples than quantitative methods, and for a study such as this, I would be comfortable seeing a sample size of 13 pig vets – provided the authors are satisfied they had data saturation in the vet sample (which it seems has been achieved, lines 173-177). Given the nature of the research question (focused on the vets’ behaviour) and the questions posed to farmers, my thoughts are that merging these two datasets is not the most appropriate course of action – and that the study would demonstrate more rigour if just the data from the vet interviews were analysed (and quotes from farmers in the results would need to be removed and replaced with suitable quotes from the vets)
Lines 98-106 – introduction of the TDF
· Technically, the TDF is a framework outlining what drives ‘behaviour’, rather than what drives ‘behaviour change’ (although I acknowledge that in the literature both these descriptions have been used and are a source of confusion!). Please can the authors review here and throughout the manuscript to ensure the TDF is referred to as a framework that outlines theoretical determinants of behaviour (rather than behaviour change).
· Line 99 - Please use another word instead of ‘universal’ (or just delete, and refer to a “comprehensive framework”). No psychological process or framework is truly universal. Perhaps the authors mean that the TDF can be used to examine any target behaviour, and is not specific to a particular type of behaviour, and is therefore usefully for applying across a range of behaviours?
· Lines 101-103 – one of the key reasons for the TDF being developed was to synthesise multiple existing psychological theories about the drivers of behaviour into one more comprehensive framework. As this is a veterinary medicine paper, and Antibiotics is an interdisciplinary journal, please can the authors provide a bit more information about the development of the TDF (only a sentence or two) to help inform those readers who will be less familiar with psychological models of behaviour (e.g. with reference to Michie et al 2005, Making psychological theory useful…; Cane et al 2012, Validation of the TDF… ; Atkins et al 2017, A guide to using the TDF… ); i.e. that the TDF synthesis multiple earlier theories, proposes 12 (v1) or 14 (v2) domains of determinants of behaviour, and perhaps provide a few examples of the domains and what sits within them.
There are two versions of the TDF - v1 (Michie et al 2005) and v2 (Cane et al 2012) – the authors need to clearly state in the methods and rest of the manuscript whether they have used TDF v1 (12 domains; Michie et al 2005) or TDF v2 (14 domains; Cane et al 2012) – as there is some confusion for the reader in terms of the references used and the names of the domains as presented in the results. At first, I assumed v2 had been used, but upon reading the results, I suspect v1 has been used – either is fine, but this needs to be made explicit. As per Atkins et al 2017, both versions are still considered acceptable for use.
Lines 187-194 – this paragraph about how the TDF coding was conducted is not fully clear to me and needs further explanation
· Lines 188-189 - The sentence beginning “The TDF domains were used to explore…” does not make sense to me, I am not clear what the authors are trying to convey. Perhaps the authors mean that the TDF framework was used to “make sense of and synthesise the 537 codes into a set of higher-order theoretical determinants of behaviour”
· Lines 189-191 – it is not clear why the theoretical determinants were ‘customized’ – and I don’t quite understand what is meant by the sentence that begins “If a code did not fit a TDF domain…”. What do the authors mean by “adequacy of the behavioural specification”? Is this a reference to their coding? In my understanding the phrase ‘behavioural specification’ usually means something about how well the target behaviour has been defined (or not). For clarity, my concern with these lines is this – have the authors adjusted the TDF domains in order to fit the coding/data? I would suggest that customizing the TDF domains is not the most appropriate approach. Instead, a better approach would be to fit to the TDF all codes that do map to the domains – and then codes that do not map can still be presented, but can be highlighted as being “additional determinants not covered by the TDF”. Examples of such codes (and supporting quotes) would need to be reported to support this. The TDF does have its limitations and doesn’t always cover everything, but I am not sure it is correct to adjust the previously validated domains of the framework – rather, it would be more useful to highlight what aspects of the dataset do not fit the TDF. If the TDF does not fully explain this behaviour, then that in itself is interesting. However, I would caution that the TDF is fairly comprehensive, so any coding which falls outside of a TDF domain would need to be clearly conceptually distinct from all the TDF domains. Please can the authors provide more information to be explicit about what they did in terms of mapping the 537 iterative codes to the TDF domains, why they made those decisions, and provide some worked examples. Or, review their coding and analysis to match the TDF domains, plus anything that does not fit.
· Lines 191-193 – what is meant by “wider contextual information… was saved separately”? Is this referring to other codes from the interviews? And were these codes included in the analysis or not? If not, why not? Surely this contextual information would also be a determinant of behaviour, and (without knowing any examples) might be relevant under the Environmental Context and Resources domain? Please can the authors clarify.
Results
Please note – I have reviewed and commented on the results on the assumption that the authors have used TDFv1 (Michie et al, 2005) – but some of my comments would be different if TDFv2 (Cane et al 2012) has been used.
TDF Terminology – throughout the results (and rest of manuscript), please use wording of TDF, rather than the adjusted wording (unless the authors can fully justify their adjustments to domains in the methods). For example, ‘Confidence in capabilities’ should be ‘Beliefs about capabilities’, ‘Emotion regulation’ should be ‘Emotion’ and Environmental influences should be “Environmental context and resources’. This is useful to avoid any confusion for the reader, and may be more helpful for future evidence synthesis (e.g. reviews of studies that use the TDF), as the evidence base grows in this area.
Grouping of determinants as internal/external
· Lines 196 – 206 – I disagree with some of the statements that support the authors’ rationale for grouping determinants as ‘internal’ and ‘external’ in lines 202-205. I do see where the authors are coming from in grouping the determinants this way (things that are about the individual’s psychological processes versus wider social and environmental influences), but strictly speaking, even those determinants that the authors have grouped as ‘internal’ are not always “influenced by their own control”. Much of these processes/determinants happen without conscious awareness and we often have much less control over these determinants than we think. If the authors wish to retain this internal/external grouping then they will need to amend the rationale to avoiding any inference that internal psychological processes are under more control than external social and environmental factors.
· Alternatively, the authors may wish to link these TDF domains to the COM-B model of behaviour (Michie et al., Implementation Science 2011, 6:42), which is a behavioural science framework that links to the TDF. Determinants (domains) within the TDF are mapped to 6 higher-order drivers of behaviour (physical capability, psychological capability, physical opportunity, social opportunity, reflective (conscious) motivation, and automatic (unconscious) motivation). Michie et al (2011) have mapped the TDF v2 to the COM-B (TDFv1 doesn’t map quite as neatly, but it can still be done). The potential benefit is that the COM-B can then be used to help identify drivers of behaviour that can be targeted in theoretically-grounded behaviour change interventions (by using the Behaviour Change Taxonomy, Michie et al., Annals of Behavioural Medicine 2013, 46:81-95).
Line 197 – ‘Conscious’ would be a better word than ‘intentional’ here
Lines 207-210 – This information about interview duration etc, would be better presented in the methods (e.g. under Study Design)
Table 3 – this is a useful table and demonstrates how the TDF domains apply to the specific behaviour of (not) adhering to the S. suis guidelines. As per my earlier suggestion, I would recommend that the domains are not grouped as internal/external, but just presented as the 12 original domains from the TDF V1.
The domain ‘Misperception’ – I am not convinced by the inclusion of this domain in the results. The authors have not presented any data extracts or arguments to support the creation and inclusion of this domain, which is not a domain in either TDF v1 or TDF v2. Further, ‘misperceptions’ as a concept are arguably beliefs about capabilities and/or consequences (e.g. beliefs that farmers will / will not do what they are asked), a reflection of social influences, and/or (poor) knowledge. Based on the current draft of the manuscript I do not think there is sufficient justification for the domain of Misperception and this domain should be removed. I understand that this will require re-working of the analysis to review and re-map codes that have been mapped to this domain, but it will improve the theoretical coherence of the results.
The TDF domain Behavioural regulation is missing from the results and analysis – I assume that nothing was coded to this domain, and it has been replaced by the authors with the domain ‘misperception’. However, it would be better for the authors to simply comment that nothing was coded to Behavioural regulation, rather than excluding it from the framework presented in the manuscript. Please can the authors be explicit about how coding did/did not map to this domain, and then amend the manuscript accordingly.
Lines 213-219 – I understand the authors’ rationale for only presenting a detailed discussion of the 6 domains identified across all the vet interviews – but it would be interesting to know how many interviews the other domains were coded across, e.g., were each of these domains mentioned by only one vet or maybe 6 or 7? Please could the authors include this information somewhere – perhaps as an additional column in Table 3 “number of interviews in which domain was coded”
Related to this, unless word count is a major issue, I would recommend that the authors consider discussing the other domains in the results in the main manuscript. Even if some domains are only mentioned by a few vets, they could still be interesting and potentially relevant determinants of behaviour for those vets – indicating that perhaps what determines behaviour for one vet might be different to the next vet. These domains could be discussed in more detail than currently presented in Table B1 and supporting quotes would help illustrate vets’ views and enhance the results.
Table B1 – if the authors choose to only present 6 domains in the main manuscript and retain an overview of the other 6 domains in the appendix, then Table B1 needs more work to ensure conceptual coherence throughout, as well as provide supporting evidence from the dataset:
· Memory, attention and decision processes – I am not convinced by the last sentence in this table cell – how can we be sure that a fast response from a vet means that advice is not forgotten? I don’t think this claim stands, unless there are quotes that can support this.
· Memory, attention and decision processes – I am not sure that the examples discussed here are really about memory, attention and decisions – they might be more about professional role. I think more detail is required here to provide clarity that these examples are conceptually coherent with the domain – or these examples could be moved to the Social/Professional Role & Identity domain.
· Nature of the behaviour – “veterinarians who are curious…” I am not convinced this is about nature of the behaviour – I would think this is more about motivations
· (Mis)perception – as mentioned above, I think this domain should be removed, and the examples presented here re-coded within existing TDF domains. I do agree that these are misperceptions, and that they are important – but I do think these can be coded elsewhere in the TDF. For example, the vet who thinks all their colleagues prefer amoxicillin – this is a social influence, it is a belief about social norms in their practice. The farmer example is more about beliefs about consequences – that by recommending amoxicillin, farmers will be satisfied.
· For all of these 6 domains (5 if misperception is deleted), the authors should add some illustrative quotes from vets to this table).
· The descriptions of the domains as relevant to the dataset could be a bit longer to provide more insight for the reader
Lines 238 – 249, Knowledge – please can the authors review the material presented here to ensure coherence with the domain. While some of the results here relate to the domain of Knowledge (e.g. not knowing the guidelines refer to sick pigs rather than dead pigs), the other statements and quotes in these two paragraphs seem to be more about the domain of Memory, Attention & Decision Processes (e.g. “respondents were not able to recall…” / “they could not remember…”)
Lines 250 – 264, Skills – the authors mostly focus on the skill of completing reports, but there are surely other skills discussed by vets (as per the discussion, e.g. making a diagnosis, communication skills). In the opening line, the authors state “the respondents described a range of skills…” – please can the authors add some information about other skills that vets talked about and include some illustrative quotes, to increase the richness of this part of the results.
Lines 289-298, Beliefs about consequences – these sentences (beginning “the veterinarians differed clearly…”) are partly about beliefs about consequences, but some of it seems to me more about social/professional role and identity – vets might hold different beliefs about the efficacy of different antibiotics but as currently drafted, these statements seem more about professional autonomy in choosing which class of drug to prescribe, and whether they choose to use 1st or 2nd choice drugs first. In the final paragraph of this section (lines 295-311) it becomes clear that the authors are discussing beliefs about consequences again, but I think these 2 paragraphs need some editing to be clearer about the conceptual coherence – and perhaps some of this could go into a section on Social/Professional Role and Identity, with additional supporting quotes.
Line 313 – do the authors mean ‘self-esteem’ or ‘self-efficacy’?
Lines 325-326 – “also mentioned the positive effect of having good colleagues and a structural education” – these factors (colleagues, education) are not really part of the domain Beliefs about capabilities – education might be more about knowledge or skills (and I suppose feeds into beliefs, e.g. “I believe I’m capable because I had a good education” – perhaps a quote here might support inclusion of this comment in this domain) – having good colleagues is really about social influences. Please can the authors review this statement and edit accordingly to ensure theoretical coherence in this domain.
Lines 350-357 – this paragraph is not really about Environmental Context & Resources. It also refers to the farmer interviews. I would suggest this paragraph be deleted.
Lines 366-368 – please can the authors also relate this statement (about relationships with feed advisers) back to the Social Influences domain.
Lines 373-375 – please can the authors also relate this statement (about professional autonomy & guidelines) back to the Social/Professional Role & Identity domain.
Discussion
The discussion is generally well-written - it neatly draws links to wider literature and mostly makes recommendations based on findings. There are some areas that do need to be reviewed by the authors.
Lines 384 – 388 – in this opening paragraph, it would be good to remind the reader that the determinants come from the TDF. Perhaps at the end of the first sentence add something about “ …S. suis guideline, using a psychological framework of behavioural determinants, the TDF (refs).”
Lines 384 – 388 – in this paragraph, it would also be good to summarise that as well as the 6 domains identified across all interviews, another 5/6 domains were also identified across some interviews (and list those as well).
Lines 392-395 – this did come out in the results, that some vets have negative perceptions about guidelines – but what is interesting here is how this is perceived as a threat to their professional autonomy (which relates to the Social/Professional Role & Identity domain). Please can the authors make the link to this domain more explicit, in both the results and the discussion.
Lines 429 – 552 – although these discussions about these TDF domains read well, and I have no major critiques as such, in places there are sentences that read more like results (e.g. the authors are summarising findings from their dataset in ways that would be more appropriate in the results sections). There are also some instances where the authors have made statements about what participants have said that have not been presented in the results. Please can the authors closely review the content of these sections from the discussion and compare with the results sections for each domain. I’m not sure much needs to be removed from the discussion, but all findings discussed do need to be added to the results. A couple of specific examples are:
· Lines 443-445 – discussion of skills, including diagnosing, administrative, communication and advisory skills – these have not been explicitly discussed in results, and would make the discussion of skills in the results much richer, especially with supporting quotes from vets.
· Lines 470-475 – discussion of vets with negative experiences of inspection authorities – needs to be discussed in the results.
Line 482-483 – This final sentence does not follow as a recommendation from the rest of the discussion about beliefs about consequences. A more theoretically appropriate intervention might be to find ways to change the beliefs of those vets who hold negative expectations and beliefs about the consequences of following S. suis guidelines, perhaps by sharing case studies from colleagues who have had positive experiences.
Line 485 – is there evidence in the dataset that these vets were ‘outspoken’? I would suggest removing this word, or adding a relevant quote to the results to support this description.
Lines 484-504 – from my own research and the wider literature I can agree that it seems likely there may be a difference in attitudes towards paperwork and guidelines between vets who have been in practice for longer, compared to those who qualified more recently – but I suspect it is not a perfect correlation. There are almost certainly some newer vets who are sceptical of guidelines and some very experienced vets who are supportive of them. I have two comments for the authors to address here please:
· It might be better to refer to these two groups of vets as “qualified for longer versus those more recently qualified” instead of older and younger vets (as age is not the same as length of time in practice) – please also check use of these terms elsewhere in the manuscript
· This is a small qualitative study, so be wary (here and in the results) of making claims about all older / all younger vets. Was this relationship between views on guidelines and paperwork the same for all of the vets who had been practicing for a shorter or longer time, or just most of them? Please either amend to say “most vets with longer experience…” or make it clear that there was a very clear relationship in this sample, but that this might not hold true across all (pig) vets.
Can the authors also add a concluding sentence or two to this section about beliefs about capabilities, recommending that intervening to boost self-efficacy amongst vets might help them adhere to the guidelines?
Line 509 – “social determinants” might be a better phrase than “causative factors”
Line 535 “as also confirmed in our results” – data about this have not been presented. Please remove this statement (or add to results, if retaining farmer data in the analysis).
Line 537 – “better conditions” – please expand this sentence to be clearer for the reader – do the authors mean better conditions in terms of biosecurity, quality of the feed etc? The authors may also wish to highlight (as indicated in lines 532-533) that some of the environmental determinants are beyond the control of both the vet and the farmer, and may require further intervention from actors such as legislators, inspectors and others.
Lines 542-552 – this section seems more like social influence and/or social/professional role & identity, as this is about the influence of (professional) peers.
As well as the discussion of the 6 domains discussed in the results, it would still be helpful to have a brief discussion of the other 6 domains (perhaps grouped together, and discussed in less depth) as these domains of determinants might still be potentially useful targets for future behaviour change interventions.
Lines 553-562 – I agree with the authors that there need to be theory-led and evidence-based strategies to support the successful implementation of guidelines. However, the authors could strengthen this paragraph by more explicitly referring to their use of the TDF as an example of this – i.e. they have used a theory-based framework to understand potential psychological, social and environmental determinants of behaviour, which means these findings can be used to identify theoretically-appropriate behaviour change techniques to support guideline implementation.
Lines 553-562 – this paragraph needs to include references to the relevant models and frameworks used, and also explain why these psychological/behavioural science frameworks are useful. Please include explicit reference to the TDF. The authors also mention behaviour change techniques, so please consider including reference to the Behaviour Change Technique Taxonomy (Michie et al., Annals of Behavioural Medicine 2013, 46:81-95) and explain how the TDF can be mapped, via the COM-B, to the BCT taxonomy.
Further, if the authors decide to link the TDF domains to the COM-B, they could frame their recommendations in the discussion with reference to the 6 core components of COM-B. For an example of using TDF, and then COM-B to map and make recommendations about to antibiotic prescribing in human medicine, the authors could review Fleming et al, 2014: https://www.ncbi.nlm.nih.gov/pmc/articles/PMC4225237/
Conclusion
Lines 586-589 – these recommendations made in the conclusions do not really follow from the findings of the study – it might be better to frame these recommendations with reference to the domains of the TDF (i.e., more explicitly draw the link between the study findings, the domains that were important and why this might suggest these recommendations)
Lines 593 – 598 – again, these recommendations (while I agree they are sensible) do not follow from the findings or the discussion of the findings. I would like to see the authors re-work this second paragraph of the conclusion to be more grounded in the findings and discussion of findings, making recommendations that link to the suggested interventions they draw out for the 6 domains in the discussion.
Line 599-600 – I agree that peer learning is likely to be useful, but based on the current drafting of the manuscript the evidence for this from this study is not clear enough – the authors could review this by ensuring they have explicitly discussed the benefits of peer learning for implementing guidelines in the results (if there are data to support this) and then linking this to the part of the discussion where they relate the study’s findings to evidence from human medicine about peer learning (which as mentioned previously should be more explicitly tied to the social influences domain of the TDF)

Reviewer 3 Report
|
Round 2
Reviewer 2 Report
I would like to thank the authors for their responses and their work to edit the manuscript, which is now clearer, and I accept the authors’ position where they have argued against making changes.
I am supportive of the manuscript being published in its current form – with two very minor changes that relate to the decision to retain the farmer data.
1. Please can the authors add back what was Table 2 in the original submission (demographic information about the farmers), and then also attribute the quotes to F1, F2, etc. Although the presentation of the farmer analysis has been edited in the results, and the emphasis is now on the vet interviews (which now also nicely emphasises the misperceptions between vets and farmers), this information about the farmer participants is still of interest
2. Relatedly, please can the authors add back the Farmer Interview Schedule into the Appendix (Table A2 in the original submission)
I also spotted that a sentence has been repeated twice in lines 598-600: “The TDF appeared a useful tool to structure the results of the interviews.”.
This study will make an important contribution to the psychological and behavioural literature relating to our understanding of antibiotic use and infectious disease management in livestock. I congratulate the authors on their efforts, and look forward to reading about the follow-up work they have mentioned making use of Intervention Mapping.
Reviewer 3 Report
The article has improved wiht the changes made by the authors.
Author Response
We thank Reviewer 3 for his/her efforts to review our manuscript again and the positive feedback about the changes we made in this version.